# Efficient and Stable *O*-Methylation of Catechol with Dimethyl Carbonate over Aluminophosphate Catalysts

**Baoqin Wu, Yao Sheng *** , **Linkai Zhou, Runduo Hong, Lifan Zhang, Xinfeng Ren, Xiujing Zou, Xingfu Shang, Xionggang Lu** and **Xueguang Wang ***

State Key Laboratory of Advanced Special Steel, School of Materials Science and Engineering, Shanghai University, No.99 Shangda Road, BaoShan District, Shanghai 200444, China
* Correspondence: shengyao@shu.edu.cn (Y.S.); wxg228@shu.edu.cn (X.W.)

**Abstract:** The *O*-methylation of catechol is an effective method for the industrial production of guaiacol used as an important chemical. However, the low catechol conversion and poor catalyst stability are the most critical issues that need to be addressed. Herein, the *O*-methylation of catechol with dimethyl carbonate was investigated over aluminophosphate (APO) catalysts, using a continuous-flow system to produce guaiacol. APO catalysts were synthesized with varying P/Al molar ratios and calcination temperatures to study their effects on catalytic performance for the reaction. The physico-chemical properties of the APO catalysts were thoroughly investigated using XRD, $NH_3$-TPD, $CO_2$-TPD, FTIR, and Py-FTIR. The P/Al molar ratio and catalyst calcination temperature significantly influenced the structure and texture, as well as the surface acid-base properties of APO. Both the medium acid and medium base sites were observed over APO catalysts, and the Lewis acid sites acted as the main active sites. The APO (P/Al = 0.7) exhibited the highest catalytic activity and excellent stability, due to the suitable medium acid-base pairs.

**Keywords:** catechol; dimethyl carbonate; *O*-methylation; aluminophosphate; guaiacol

## 1. Introduction

It is well-known that guaiacol is a vital chemical feedstock and intermediate for the production of flavorings, fragrance, pharmaceuticals and a variety of other specialty chemicals [1–4]. Traditionally, the *O*-methylation of catechol with toxic alkylating agents (methyl halide, methanol, phosgene or dimethyl sulfate) using a stoichiometric amount of strong base is usually adopted for the industrial production of guaiacol, but this method produces large amounts of hypersaline organic wastewater and the stoichiometric consumption of NaOH [4–6]. Dimethyl carbonate (DMC), a green methylating reagent, is considered as a nontoxic, efficient, and environmentally acceptable alternative to toxic methylation reagents [7–10]. Tekale and coworkers performed the *O*-methylation of catechol with DMC, using ionic liquid as a catalyst with high *O*-selectivity [11]. However, considering the difficulty of catalyst recovery and product separation in the homogeneous catalytic system, the development of an environmentally benign heterogeneous catalyst for the *O*-methylation of catechol with DMC, attracts significant interest.

A variety of solid acid-base catalysts, including Mg–Al hydrotalcite [12,13], alkali metal ion-loaded MgO [14], $AlPO_4$–$Al_2O_3$ [15], Trihexyl (tetradecyl)-phosphonium bromide [16], and $Zn_{1-x}Mg_xO$ nano-spheres [17] have been investigated for the *O*-methylation of catechol with DMC. Among the above catalysts, aluminophosphate catalysts have the advantages of thermal stability and tunable acidity and baseity, which are closely related to the adsorption of reactant molecules and catalytic activity. In general, it is well-known that the P/Al molar ratio of aluminophosphate catalysts is a critical parameter for adjusting the acid-base property, which greatly influences the catalytic performance [18–22]. According to the literature reports, a variety of phosphate-treatment methods have been conducted on

aluminophosphate catalysts to change the phase composition and acid-base properties, such as silicoaluminophophates (SAPOs), metalalumimophosphates (MAPOs) and transition metal phosphates, morphous aluminophosphate (AlP) and metal-alumino-phosphates (MAlPs) [23–31]. These processes demonstrate that the synthesis of aluminophosphate catalysts with an adjustable P/Al ratio can be realized by pursuing novel approaches.

In the present work, aluminophosphate (APO) catalysts with different P/Al molar ratios and calcined temperatures were synthesized using the sol-gel method for the selective *O*-methylation of catechol with DMC. The effect of the P/Al molar ratios and calcined temperatures on the structure, acid-base properties and catalytic performance were investigated using multiple techniques, including XRD, FT-IR, SEM, $CO_2$-TPD, and $NH_3$-TPD. The APO (P/Al = 0.7) that was calcined at 475 °C exhibited high activity for the vapor-phase synthesis of guaiacol from catechol with DMC. The stability of the APO catalysts was examined through catalytic as well as characterization studies.

## 2. Results and Discussion

### 2.1. Characterization of APO Catalysts

The XRD patterns of APO catalysts with various P/Al ratios and calcination temperatures are shown in Figure 1. APO(0.6)-475 and APO(0.7)-475 exhibited a very broad and weak peak in the 2θ range of 15−30° (Figure 1a), which was a feature peak of amorphous $AlPO_4$ (PDF#20-0045) [32,33]. As the P/Al molar ratio increased in the APO samples, no other characteristic diffraction peaks were observed in the XRD pattern, but the peak intensity of the $AlPO_4$ became much stronger and sharper, indicating the partial crystallization of the AlPO framework on the sample with relatively high P/Al ratios ($\geq$0.8). Figure 1b illustrates the XRD patterns of the APO catalysts with the P/Al ratio of 0.7, at different calcination temperatures. APO(0.7)-375 showed broad characteristic diffraction-peaks centered at 14.0°, 28.2°, 38.4°, 48.8°, and 65.0°, which was consistent with the standard AlOOH (PDF#49-0133) [34]. Notably, the peak intensity of the pseudo-boehmite became weak with the increase in calcination temperatures, which may be related to the transformation from the pseudo-boehmite crystalline phase to the aluminophosphate phase. It has been reported that the P addition facilitated the transformation of the pseudo-boehmite crystalline phase into the amorphous AlPO phase [35]. In addition, by increasing the calcination temperature from 375 to 575 °C, all samples showed no obvious change in the intensity of the amorphous $AlPO_4$, and no characteristic diffraction peaks of other oxides such as $Al_2O_3$ and $P_2O_5$ were observed, indicating that the amorphous $AlPO_4$ catalyst synthesized using the sol-gel method had good thermal stability. The real P/Al molar ratios in the prepared materials were measured by inductively coupled plasma optical emission spectrometry (ICP-OES), and the results are given in Table 1. In addition, as seen in Table 1, all the catalysts showed a similar specific-surface-area, except the APO(0.9)-475.

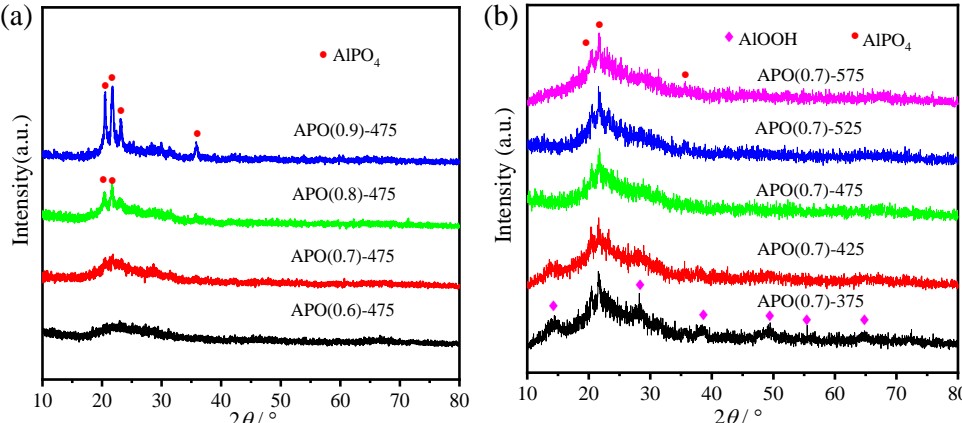

**Figure 1.** XRD patterns of APO catalysts with different (**a**) P/Al ratios and (**b**) calcination temperatures.

**Table 1.** Physico-chemical characteristics of APO catalysts.

| Sample | Specific Surface Area (m² g⁻¹) [a] | P/Al Molar Ratio [b] |
|---|---|---|
| APO(0.6)-475 | 116 | 0.58 |
| APO(0.7)-475 | 120 | 0.66 |
| APO(0.8)-475 | 125 | 0.79 |
| APO(0.9)-475 | 35 | 0.85 |
| APO(0.7)-375 | 124 | 0.71 |
| APO(0.7)-425 | 123 | 0.68 |
| APO(0.7)-525 | 117 | 0.65 |
| APO(0.7)-575 | 116 | 0.63 |

[a] Calculated using the Brunauer–Emmett–Teller (BET) method. [b] Obtained by ICP.

We subsequently utilized FT-IR to confirm the structures of aluminum phosphate catalysts. The FT-IR spectra of APO catalysts with different P/Al molar ratios and calcination temperatures are shown in Figure 2. The broad absorption band at 1641 cm⁻¹ was ascribed to the stretching and bending vibrations of the -OH groups, which may be associated with the physisorption of water molecules or surface hydroxyls on the catalysts [36,37]. The strong adsorption band at around 1125 cm⁻¹ was ascribed to an asymmetric P-O-Al stretching vibration, and the one at 488 cm⁻¹ was assigned to symmetrical P-O-Al deformation vibration [38,39]; the slight redshift of the band observed with a higher P/Al ratio and calcination temperature may be due to small structural differences. In addition, the IR spectra showed a weak peak at 730 cm⁻¹, which was assigned to the symmetric stretching mode of the P–O–P bond [40]. The XRD and FT-IR analyses suggested the formation of the AlPO₄. When the P/Al molar ratios increased from 0.6 to 0.9, the intensity of the characteristic peaks at 1125, 730, and 488 cm⁻¹ of the APO catalysts increased significantly, while the intensity of these peaks for the APO catalysts with different calcination temperatures remained essentially unchanged, indicating that the crystallinity of the AlPO₄ phase increased with the P/Al molar ratios, which was consistent with the XRD.

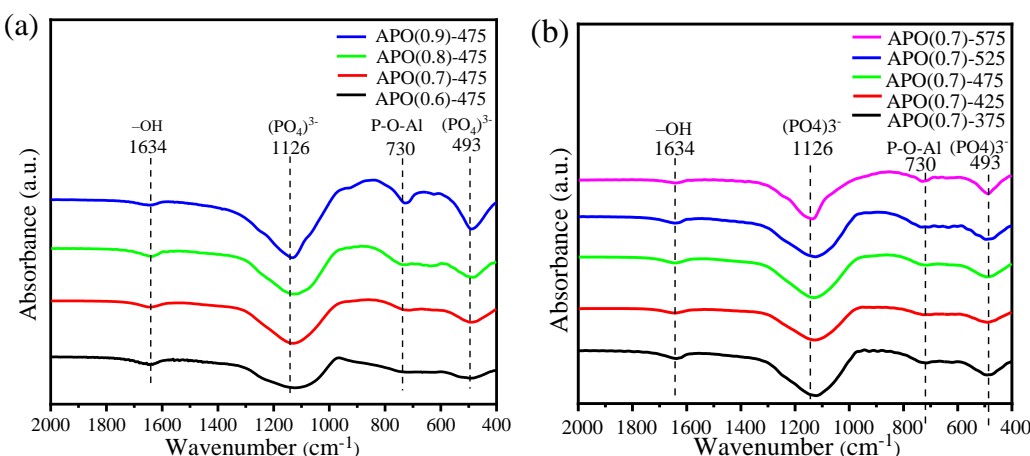

**Figure 2.** FT-IR spectra of APO catalysts with different (**a**) P/Al molar ratios and (**b**) calcination temperatures.

The surface acidity and baseity of the APO catalysts were measured using the NH₃-TPD and CO₂-TPD techniques, as shown in Figure 3. For the NH₃-TPD profiles (Figure 3a,b), all the samples showed one broad desorption peak in the range of 50−300 °C, which was typical for amorphous materials [41]. The broad desorption peak could be deconvoluted into two peaks with maximal temperatures in the region of 100–150 °C and 170–320 °C, corresponding to the weak and medium acid-sites (Al³⁺), respectively [42]. Notably, it was revealed that the total amount of weak and medium acidity decreased with the increment in the P/Al molar ratios, whereas the number of weak acid-sites increased slightly at first and then decreased significantly with the enhanced P/Al molar ratios, as shown in Figure 3a.

Therefore, the APO(0.7)-475 had the largest amount of medium acid-sites, compared with other APO catalysts. Before adding phosphoric acid, the catalyst preparation system was a γ-AlOOH sol formed by peptizing pseudoboehmite and nitric acid. The addition of phosphoric acid was equivalent to introducing the P element into the Al-O bond system; and thus, an Al-O-P covalent bond formed. As the P/Al molar ratios increased, the covalent bond of Al-O-P increased, the acid-base properties of the APO catalyst changed. The $NH_3$-TPD profiles of APO catalysts with different calcination temperatures are presented in Figure 3b. When the calcination temperature increased from 375 °C to 575 °C, the acidity and the number of acid sites did not change significantly, indicating that the catalyst had a high calcination-temperature range and good thermal stability.

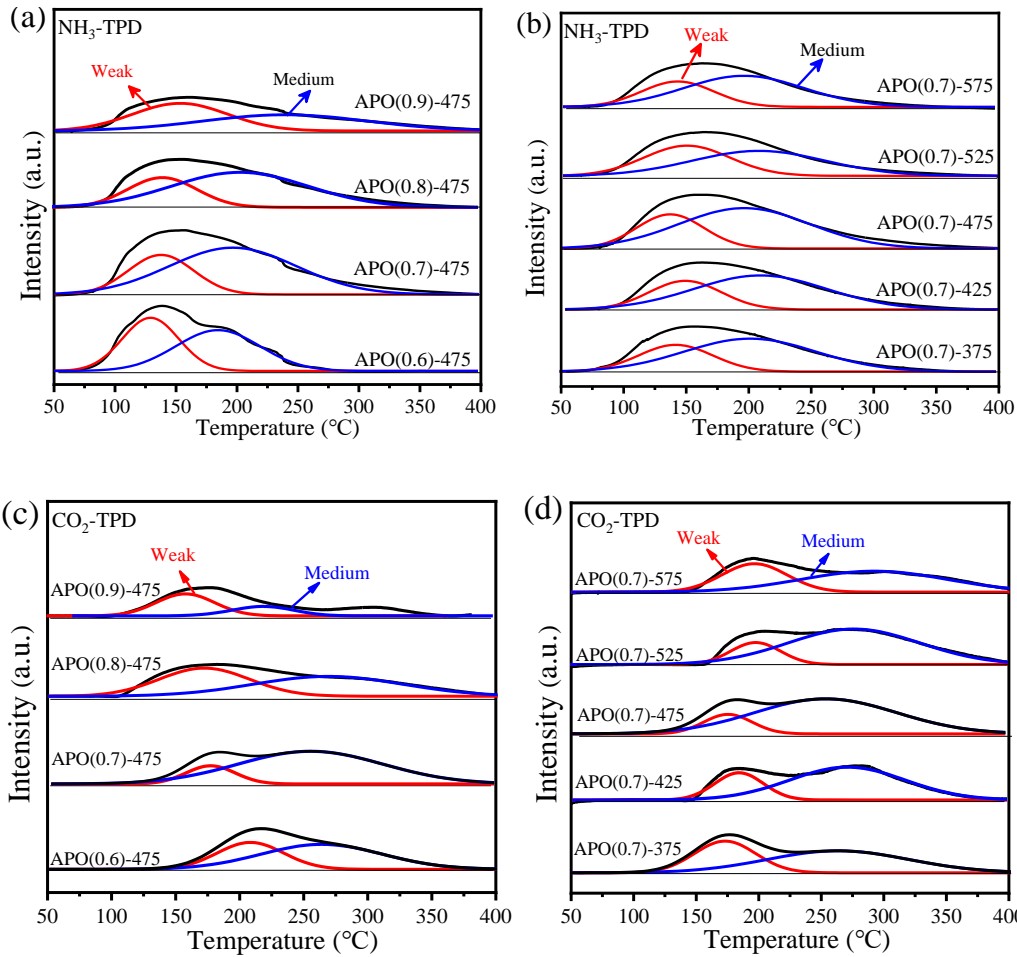

**Figure 3.** TPD profiles of (**a**,**b**) $NH_3$-TPD and (**c**,**d**) $CO_2$-TPD of APO catalysts with different P/Al ratios and calcination temperatures.

In the case of the $CO_2$-TPD profiles (Figure 3c,d), all the catalysts appeared to have two $CO_2$ desorption peaks, corresponding to weak (below 200 °C) and medium/strong (250−300 °C) base sites. The weak and medium base sites were derived from adsorption at the surface Brönsted base sites ($OH^-$ groups) and the metal-oxygen pair ($Al^{3+}$–$O^{2-}$), respectively [43,44]. In the range of P/Al molar ratios of 0.60 to 0.90 (Figure 3c), the peak temperatures of the $CO_2$ desorption peak all moved to a low temperature with the increase in the P/Al ratio, indicating that the solid-base strengths were decreased. However, the number of medium base sites increased first and then decreased with enhanced P/Al molar ratios; thus, the APO(0.7)-375 had the largest number of medium base sites, compared with other APO catalysts. Figure 3d shows the $CO_2$-TPD profiles of the APO catalysts with different calcination temperatures. Notably, it was revealed that all concentrations of the medium base sites increased first and then dropped slightly with the increase in calcination

temperature, whereas the total amount of weak and medium acid sites decreased with the enhanced calcination temperature. Thus, the APO(0.7)-475, with a calcination temperature of 475 °C had a relatively large density of the medium base sites. It has been reported that the medium acid sites are associated with the activation of carbonyl oxygen of DMC, while the medium base sites (mostly $Al^{3+}$–$O^{2-}$) act as active centers for activating phenol, to produce phenoxyl species [45,46]. The acid sites alone or the base sites alone on the catalyst surface cannot simply be the dominant factor enhancing catalyst performance, since the physically mixed solid-acid and base catalyst also exhibited lower activity.

We further performed the pyridine-adsorbed FT-IR (Py-FTIR) to distinguish between the Brönsted and Lewis acid sites present in the solid-acid catalyst, and spectra are presented in Figure 4. All APO catalysts exhibited three FT-IR bands in the fingerprint region of 1400–1700 cm$^{-1}$. According to the literature, two significant bands at around 1614 and 1449 cm$^{-1}$ were assigned to the pyridine adsorbed on coordinatively unsaturated aluminum-ions and Al-O-P (L) [47], indicating that the catalyst surface was predominantly covered by Lewis acid sites. In addition, the band at 1492 cm$^{-1}$ was assigned to the adsorption of the pyridine molecule on both the Brönsted and the Lewis acid sites (B + L) [48], indicating that both Brönsted and Lewis acid sites existed in all samples. It was obvious from the Figure 4 that all samples exhibited a higher amount of Lewis acid sites compared to Brönsted acid sites. The intensity of the band at 1448 cm$^{-1}$ increased with the P/Al ratio increasing from 0.6 to 0.7, which could be attributed to the increase in the phosphorus content. However, a further increasing P/Al ratio reduced the total number of acid sites, indicating that the surface Lewis acidity of the APO catalysts was weakened, which was consistent with the NH$_3$-TPD results. Therefore, we can conclude that the P/Al molar ratio had a significant effect on the Lewis acid sites, and the total acid concentration could be readily adjusted via the application of catalyst-component modification.

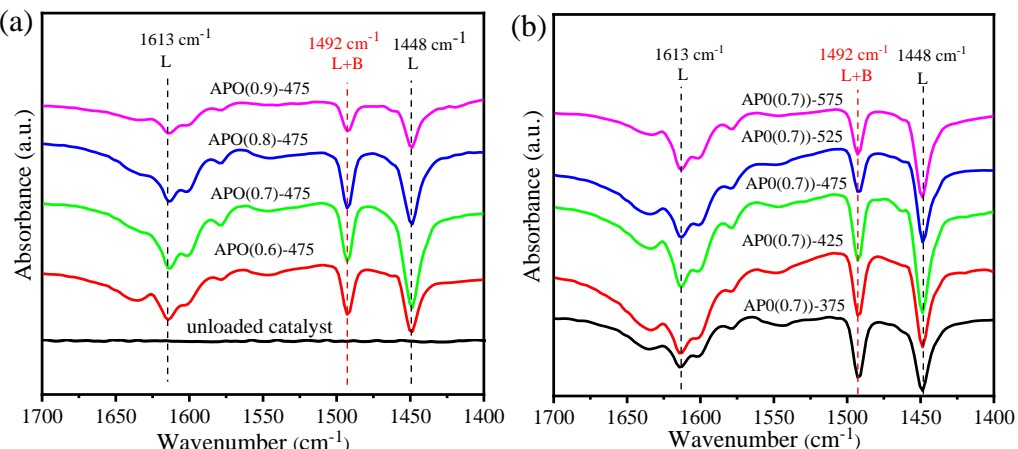

**Figure 4.** FT-IR spectra after pyridine adsorption of APO catalysts with (**a**) different P/Al ratios and (**b**) calcination temperatures.

The Py-FTIR spectra of APO catalysts with different calcination temperatures are shown in Figure 4b. All samples had prominent characteristic peaks at 1614, 1578, 1492, and 1449 cm$^{-1}$, indicating the existence of both L and B acid sites. An additional weak band observed at 1545 cm$^{-1}$ can be attributed to pyridine adsorbed on the B acid sites, indicating that the APO catalysts contained relatively few B acid sites. When the calcination temperature increased from 375 to 475 °C, the L site concentration increased, while its influence on the B acid sites was negligible. Above 475 °C, the catalyst showed a decrease in the number of acid sites on both the L and the B acid. The acidity of the L acid first increased and then weakened, which may be caused by the increase at first and then the decrease in the anion vacancies on the surface of the APO catalyst [49].

### 2.2. O-Methylation of Catechol over APO Catalysts

The vapor-phase selective *O*-methylation of catechol with DMC to guaiacol using APO catalysts was investigated as a representative reaction, and the results are e shown in Figure 5. The catalytic experiments were carried out at 300 °C under solvent-free conditions, with a 1: 6 molar ratio of catechol to DMC. Figure 5a shows the catalytic data of APO catalysts with different P/Al molar ratios (calcined at 475 °C).A total of 80% catechol conversion with 70% guaiacol selectivity was obtained when the P/Al molar ratio was 0.6. When the P/Al molar ratio increased from 0.6 to 0.7, the catechol conversion increased, and reached a maximum of 94%. The APO(0.7)-475 exhibited excellent catalytic activity compared to the previously reported catalysts (Table 2). The superior catalytic activity of the APO(0.7)-475 could be explained by the presence of a higher number of acid sites associated with the medium acid sites. With the further increase in the P/Al ratio, to 0.8, the catalysts showed high catechol conversion of ~94% in the initial 1~2 h of the reaction, but the catechol conversion decreased to < 75% after 12 h. The reaction outcome then decreased considerably, and there was only 14% catechol conversion after 12 h, with a further increase in the P/Al ratio to 0.9, which may be associated with the observations that the catalyst with a high P/Al ratio ($\geq$ 0.8) had the lowest specific-surface-area and the highest crystallinity of the $AlPO_4$ phase, compared with amorphous $AlPO_4$. As is well known, the medium acidity of catalysts is responsible for the carbonyl oxygen of DMC, while the medium-strength Lewis base-sites favor the activation of phenol to produce phenoxyl species. Therefore, a catalyst containing a suitable number of both Lewis and Brønsted acid-sites promoted the *O*-methylation of catechol with DMC.

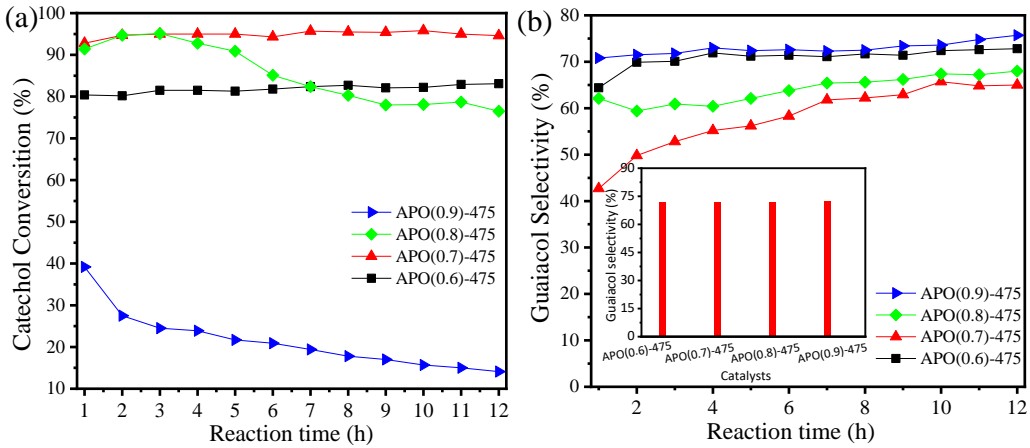

**Figure 5.** Effect of reaction time on (**a**) catechol conversion and (**b**) guaiacol selectivity over the APO catalysts with different P/Al molar ratios; the inset in (**b**) is the guaiacol selectivity at ~30% conversion of catechol. Reaction condition: 3 g APO catalyst, molar ratio of catechol to DMC = 1:6, 300 °C, liquid hourly space velocity (LHSV) = 0.4 $h^{-1}$.

**Table 2.** *O*-methylation of catechol to guaiacol over our proposed APO(0.7)-475 catalysts and recently documented catalysts.

| Catalyst | Temperature (°C) | Conversion (%) | Selectivity (%) | Ref |
|---|---|---|---|---|
| $AlP_{1.1}Zr_{0.012}$-400 | 280 | 82 | 89 | [1] |
| M-P-O | 270 | 80 | 92 | [5] |
| $La_{1-x}Ti_xPO_4$ | 275 | 75 | 79 | [6] |
| Mg–Al | 325 | 94 | 79 | [12] |
| $AlPO_4$–$Al_2O_3$ | 300 | 11.7 | 87 | [15] |
| Al-P-O | 275 | 58 | 97 | [18] |
| APO(0.7)-475 | 300 | 94 | 63 | This work |

Figure 6 shows the catalytic data of APO catalysts with different calcination temperatures (P/Al molar ratios = 0.7). When the calcination temperature increased from 375 to 475 °C, the catechol conversion increased from 89 to 95.4%, and the guaiacol selectivity decreased from 65.9 to 63.2%. Catechol conversion decreased from 95.4 to 85.1% and a substantial decline was observed at longer time on stream, with a complimentary increase in guaiacol selectivity from 63.2 to 70.1% when the calcination temperature further increased from 475 to 575 °C; this indicates the significant influence of the calcination temperature on catechol conversion within the range explored. The APO(0.7)-475 exhibited the highest catalytic activity among the other APO catalysts, due to a combined acid–base catalyzed *O*-methylation of phenol, in which the acid sites of the catalyst were associated with the carbonyl oxygen of DMC, and the base sites of the catalyst were associated with the activation of the hydroxyl group of catechol adsorbed on the surface of the catalyst.

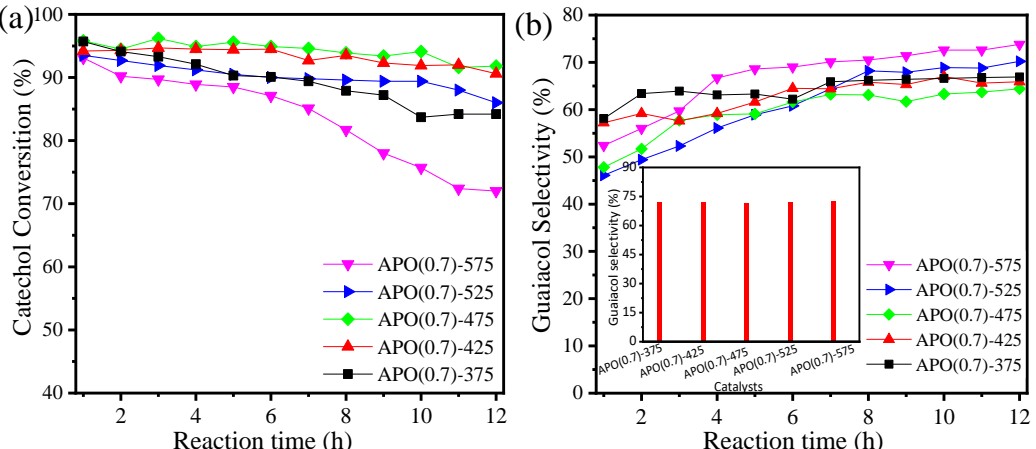

**Figure 6.** Effect of reaction time on (**a**) catechol conversion and (**b**) guaiacol selectivity over the APO catalysts with different calcination temperatures; the inset in (**b**) is the guaiacol selectivity at ~30% conversion of catechol. Reaction condition: 3 g APO catalyst, molar ratio of catechol to DMC = 1:6, 300 °C, LHSV = 0.4 h$^{-1}$.

Stability is crucial for an industrial catalyst; hence, the durability experiment of the selective *O*-methylation of catechol with DMC to guaiacol over APO catalysts was undertaken. Figure 7a shows the stability test results of the optimized APO(0.7)-475 catalyst under optimal conditions for 120 h. In the reaction time from 0 to 20 h, the conversion of catechol increased from 91.0% to 95.4%, but the selectivity of guaiacol decreased from 60.7% to 60.4% and the selectivity of veratrol decreased from 36.1% to 34.8%; after the reaction time of 20−80h, the catechol conversion was baseally maintained at approximately 95.0%, and the selectivity of guaiacol was baseally maintained in the range of 62.2−65.1%; as the reaction time increased to 120 h, the APO(0.7)-475 catalyst still maintained a relatively satisfactory activity (93.7% catechol conversion); the selectivity of guaiacol increased from 65.1 to 67.3%, and the selectivity of veratrol decreased from 32.3 to 29.7%. These results demonstrated that the catalysts had reasonable stability and could be used for long time in the flow system with high catalytic-activity and guaiacol-selectivity.

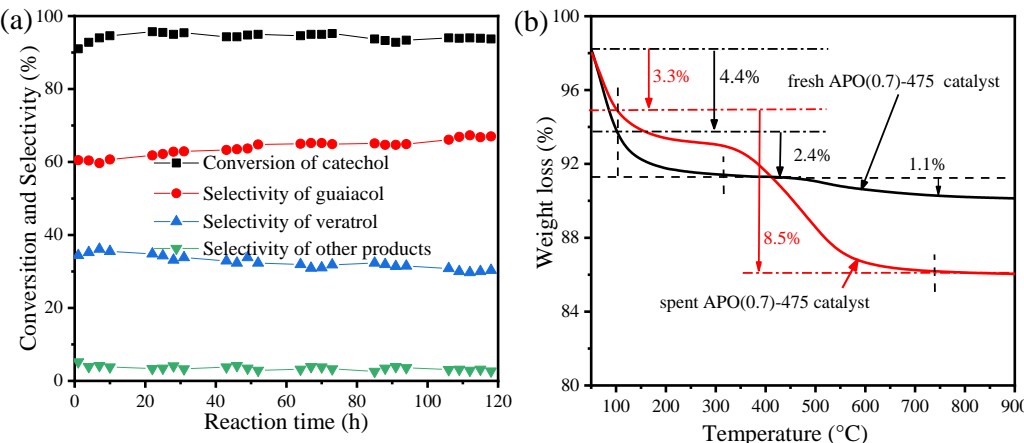

**Figure 7.** (**a**) Stability test of APO(0.7)-475 for the selective *O*-methylation of catechol with DMC to guaiacol. Reaction condition: 3 g APO catalyst, molar ratio of catechol to DMC = 1:6, 300 °C, LHSV = 0.4 h$^{-1}$. (**b**) TG profiles of the fresh and spent APO(0.7)-475.

However, there was a slight loss in catalytic activity after 120 h of reaction time, as shown in Figure 7a; therefore, the fresh and spent APO(0.7)-475 catalysts (after 120 h) were comparatively characterized. In order to study the thermal-decomposition process of the optimal catalyst and the carbon deposition during the reaction process, the fresh and spent catalysts were characterized using thermogravimetric analysis (TG). As shown in Figure 7b, the curve of the fresh APO(0.7)-475 catalyst showed a continuous weight loss from 50 to 900 °C, where three weight-loss regions were observed. The first weight loss (4.4 wt%) at a temperature lower than 100 °C was attributed to the volatilization of absorbed water. The second weight loss, approximately 2.4%, from 100 to 316 °C, suggested the dehydroxylation of hydrophosphates in conjunction with the formation of phosphates [50], while the weight loss above 475 °C was attributed to the increase in crystallized aluminum phosphate on the catalyst surface or the collapse of the pore structure. However, the spent APO(0.7)-475 catalyst showed different weight-loss behaviors, where two weight-loss regions were observed. The first weight loss, of approximately 3.3%, was due to the physical dehydration and the vaporization of low-boiling-point reagents on the catalyst surface at a temperature of 50 to 100 °C. The curve of the spent APO(0.7)-475 catalyst showed a continuous weight loss, from 100 to 682 °C, which could be divided into two sections: 100 to 312 °C, and 312 to 682 °C. The weight loss which occurred at 100 to 312 °C was assigned to the vaporization/decomposition of chemisorbed reagents and/or high-boiling-point organic compounds adsorbed on the catalyst surface. The weight loss which occurred at 312 to 682 °C was due to the decomposition of most of the carbon deposited [51].

## 3. Materials and Methods

### 3.1. Materials

Pseudo-boehmite, nitric acid (HNO$_3$), phosphoric acid (H$_3$PO$_4$), ammonium hydroxide (NH$_3$·H$_2$O), dimethyl carbonate (DMC), guaiacol, and veratrol were purchased from Sinopharm Chemical Reagent Co., Ltd. (Shanghai, China). All the reagents were commercially available and used without any further purification.

### 3.2. Catalyst Preparation

The APO catalysts were prepared using a sol-gel method. In detail, pseudo-boehmite (AlOOH) (100 g) was dispersed in 300 mL of deionized water, and the resulting solution was stirred at 60 °C for 30 min. Then 30 mL of HNO$_3$ (65 wt%) was added into the solution and a sol was formed. After stirring for 30 min, a specific amount of H$_3$PO$_4$ (85 wt%) was dropped into the solution and stirred for 60 min, and the molar ratios of P/Al were set at 0.6:1, 0.7:1, 0.8:1, and 0.9:1. Subsequently, NH$_3$·H$_2$O (28 wt%) was added slowly, followed by stirring for another 1 h, when the pH of the system was adjusted to 8 ± 0.2.

The resulting mixed colloid was then tightly closed and kept at 60 °C and, after 48 h, the gel was dried at 100 °C for 24 h. Finally, the dried solid was calcined at 375−575 °C for 12 h. The obtained catalyst was denoted as APO(x)-T (x and T referred to the molar ratio of P/Al and the calcination temperature, respectively).

### *3.3. Catalyst Characterization*

The XRD diffraction patterns of the catalyst in this paper were recorded by D8 Adance X-ray diffractometer produced by Bruker AXS, Germany. The XRD analysis of the catalysts was made in the scanning range of 2θ = 10−90°, with a scan speed of 8° min$^{-1}$, using Cu Kα radiation (λ = 0.15418 nm). The P/Al molar ratios in the catalysts were analyzed using ICP-OES conducted on a Perkin Elmer emission spectrometer, Massachusetts, USA. The specific surface area was obtained on a Micromeritics ASAP 2020 analyzer, Norcross, Georgia, USA. Temperature-programmed desorption experiments of ammonia ($NH_3$-TPD) and temperature-programmed desorption of $CO_2$ ($CO_2$-TPD) were conducted on a Biotech PCA-1200 chemisorption analyzer, China. Before $NH_3$-TPD ($CO_2$-TPD), the catalyst was pretreated under an Ar stream at 200 °C for 60 min, to remove water and other adsorbates; it was then cooled down to 50 °C, followed by saturation with $NH_3$ ($CO_2$) for 30 min. The catalyst was then heated to 700 °C at a ramping rate of 10 °C min$^{-1}$. A TCD was used to monitor the amount of $NH_3$ ($CO_2$) desorbed. The IR Spectrum was recorded on a Tensor 27 spectrometer (Bruker, Germany) covering field 400−4000 cm$^{-1}$. For sample preparation, 20 mg of the catalyst sample was mixed with 0.2 mg of KBr and pressed on a laboratory press. Pyridine-adsorbed FT-IR (Py-IR) spectra were collected on a Tensor 27 spectrometer (Bruker, Germany) after being activated at 200 °C, followed by pyridine adsorption at 50 °C. After adsorption, the catalysts were step-wise heated to 50–200 °C (step 50 °C), and the spectra were taken at each temperature. Difference spectra were obtained by subtracting the baseline spectrum from those of the sample interacting with the adsorbed molecule. Thermogravimetric analysis (TGA) curves were carried out on a Netzsch STA 449F3 thermal analyzer(NETZSCH, Germany) from 25 to 900 °C, with a heating rate of 10 °C min$^{-1}$ in a flow of air (40 mL min$^{-1}$).

### *3.4. Catalytic Activity Tests*

The methylation of catechol over the APO catalysts was carried out in a fixed-bed quartz reactor at atmospheric pressure. In a typical test, the catalyst (3.0 g, 20–40 meshes) was placed in the middle of the reactor fitted with a thermocouple for temperature measurement. Firstly, the quartz was filled with a $N_2$ atmosphere. The feedstock with the molar ratio of catechol/DMC = 1:6 was then pumped into the reactor (flow rate, 0.15 mL min$^{-1}$). The reaction mixture was then heated at 300 °C. The liquid products were analyzed off-line using gas chromatography–mass spectrometry (GC-MS) (Shimadzu GCMS-QP 2010 Plus) and GC (GC9790) with a capillary column (column HP-5, 30 m length, 0.32 mm internal diameter, 0.25 μm film thickness) connected to an FID detector.

### 4. Conclusions

In summary, the APO catalyst, prepared using the facile sol-gel method, was found to be an efficient catalyst for the selective *O*-methylation of catechol using dimethyl carbonate as the alkylating agent. The P/Al molar ratio and catalyst calcination temperature significantly influenced the structure and texture, as well as the surface acid-base properties of APO. Both the medium acid and medium base sites were observed over APO catalysts, and the Lewis acid sites acted as the main active sites. The materials had lower crystallinity with a higher medium acid-base site concentration, which facilitated the catalytic activity in the *O*-methylation of catechol to guaiacol. Under optimized reaction conditions, APO(0.7)-475 with P/Al molar ratio of 0.7 calcined at 475 °C exhibited an excellent catechol conversion of 95%. The excellent catalytic performance was due to the co-operative effect of both the medium acid sites and the base sites present in the catalyst, which facilitated the adsorption of reactant molecules and promoted the activation of carbonyl oxygen of DMC. In addi-

tion, APO(0.7)-475 was highly stable and could maintain a relatively satisfactory catechol conversion (93.7%) after a reaction time of 120 h, in the continuous-flow experiment. We believe that this simple, cost-effective, and environmentally benign catalytic process has excellent application potential in the alkylation of phenols.

**Author Contributions:** Conceptualization, B.W., Y.S. and X.W.; investigation, L.Z.(Linkai Zhou) R.H. and L.Z.(Lifan Zhang); data curation, X.R.; writing—original draft preparation, B.W. and Y.S.; writing—review and editing, X.Z. and X.S.; funding acquisition and supervision, X.L. and X.W. All authors have read and agreed to the published version of the manuscript.

**Funding:** This work was financially supported by the Science and Technology Commission of Shanghai Municipality (Grant No. 21DZ1208900) and Shanghai Engineering Research Center of Green Remanufacture of Metal Parts (No. 19DZ2252900).

**Data Availability Statement:** Not applicable.

**Acknowledgments:** The authors thank Shanghai University for supporting this work.

**Conflicts of Interest:** The authors declare no conflict of interest.

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
