# Peer review of "Efficient and Stable O-Methylation of Catechol with Dimethyl Carbonate over Aluminophosphate Catalysts"

_catalysts, doi:10.3390/catal13010150_

Round 1
Reviewer 1 Report
Journal : Catalysts
Manuscript title : Efficient and stable O-methylation of catechol with dimethyl carbonate over aluminophosphate catalysts
Manuscript ID: catalysts-2129154
Authors: Baoqin Wu, Yao Sheng, Linkai Zhou, Runduo Hong, Lifan Zhang, Xinfeng Ren, Xiujing Zou, Xingfu Shang, Xionggang Lu and Xueguang Wang
Dear Editor,
The manuscript submitted to your journal and, entitled and coded as above, deals with the synthesis of aluminophosphate (APO) and its catalytic performance in the O-methylation of catechol. The authors discussed, on the one hand, the synthesis and characterization results of APO and, on the other hand, the set-up of optimal conditions with regards to APO for a higher selectivity of guaiacol from the reaction of catechol with dimethyl carbonate.
The paper is recommended for publication after taking into account the following remarks:
1. Several flaws as to the text including typesetting and grammatical errors, and others were found and must be corrected (see the attached file for some corrections).
2. How do the authors account for the formation of 1,2-diethoxybenzene? Why was 2-ethoxyphenol (guaethol) not detected, while it is expected to be formed more predominantly than the 1,2-diethoxybenzene?
3. The authors used a catechol/DMC ratio of 1:6, that is five-fold excess of the methylating reagent. With this excess, one might expect a higher selective of veratrol over guaicol. A rationale as to the higher selectivity of guaicol is recommended.
4. The following review on O-methylation of organic compounds must be mentioned in this article:
Moulay, S. O-Methylation of Hydroxyl-Containing Organic Substrates: A Comprehensive Overview, Current Organic Chemistry, 22(20), 1986-2016, 2018.
5. The legend of Figure 7b is missing.
6. The present authors’ results were not discussed in comparison with those of other workers using other catalysts.
7. Lines 140-141: the sentence “It has been reported.....phenoxyl species” needs to be rewritten as it is devoid of meaning.
8. Line 196: “Then the reaction activity...” should be better “Then the reaction outcome...”. The phrase “and only 196 14% catechol conversion after 12 h with further increase of P/Al ratio to 0.9” requires a verb.
9. Line 315: “The synthesis of guaiacol from catechol...” should be changed to “The methylation of catechol...”.
10. In the subsection 3.1.Materials, reactants and control molecules (guaiacol, veratrol, 1,2-diethoxybenzene) should be indicated.

Author Response
- Several flaws as to the text including typesetting and grammatical errors, and others were found and must be corrected (see the attached file for some corrections).
R: Thanks a lot for your suggestions. We have checked the manuscript carefully and improved the manuscript.
- How do the authors account for the formation of 1,2-diethoxybenzene? Why was 2-ethoxyphenol (guaethol) not detected, while it is expected to be formed more predominantly than the 1,2-diethoxybenzene?
R: We are very grateful for the reviewer’s suggestions and academic rigor. We corrected “1,2-diethoxybenzene” into “veratrol” in the Figure 7a in the revised manuscript (page 9).
- The authors used a catechol/DMC ratio of 1:6, that is five-fold excess of the methylating reagent. With this excess, one might expect a higher selective of veratrol over guaicol. A rationale as to the higher selectivity of guaicol is recommended.
R: Thanks a lot for the reviewer’s suggestions. The veratrol is formed by the subsequent O-alkylation of guaiacol, and its selectivity is influenced by liquid hourly space velocity, DMC/catechol molar ratio and reaction temperature. Although the DMC/catechol molar ratio was five-fold excess of the methylating reagent, liquid hourly space velocity and reaction temperature in our work were not conducive to form veratrol.
- The following review on O-methylation of organic compounds must be mentioned in this article: Moulay, S. O-Methylation of Hydroxyl-Containing Organic Substrates: A Comprehensive Overview, Current Organic Chemistry, 22(20), 1986-2016, 2018
R: Thanks a lot for the reviewer’s suggestions. We cited this paper in the revised manuscript as ref. 10. Correspondingly, we also revised the reference number.
- The legend of Figure 7b is missing.
R: Thanks a lot for the reviewer’s suggestions. We have supplemented the legend of Figure 7b in the revised manuscript (page 9).
- The present authors’ results were not discussed in comparison with those of other workers using other catalysts.
R: Thanks a lot for the reviewer’s suggestions. We have added the following Table R1 and relevant descriptions into the revised manuscript (page 8).
Table R1. O-methylation of catechol to guaiacol over our proposed APO(0.7)-475 catalysts and recently documented catalysts.
|
Catalyst |
Temperture (oC) |
Conversion (%) |
Selectivity (%) |
Ref |
|
AlP1.1Zr0.012-400 |
280 |
82 |
89 |
1 |
|
M-P-O |
270 |
80 |
92 |
5 |
|
La1−xTixPO4 |
275 |
75 |
79 |
6 |
|
Mg–Al |
325 |
94 |
79 |
12 |
|
AlPO4–Al2O3 |
300 |
11.7 |
87 |
15 |
|
Al-P-O |
275 |
58 |
97 |
18 |
|
APO(0.7)-475 |
300 |
94 |
63 |
This work |
- Lines 140-141: the sentence “It has been reported.....phenoxyl species” needs to be rewritten as it is devoid of meaning.
R: Thanks a lot for the reviewer’s suggestions. We have re-described “It has been reported.....phenoxyl species” in the old manuscript into “It has been reported that the medium acid sites associate with the activation of carbonyl oxygen of DMC, while the medium basic sites (mostly Al3+–O2-) act as active centers to activate phenol to produce phenoxyl species” in the revised manuscript (page 4).
- Line 196: “Then the reaction activity...” should be better “Then the reaction outcome...”. The phrase “and only 196 14% catechol conversion after 12 h with further increase of P/Al ratio to 0.9” requires a verb.
R: We are very grateful for the reviewer’s suggestions and academic rigor. We have re-described “Then the reaction activity...” into “Then the reaction outcome...” and added a verb in the phrase “and only 14% catechol conversion after 12 h with further increase of P/Al ratio to 0.9” in the revised manuscript (page 6).
- Line 315: “The synthesis of guaiacol from catechol...” should be changed to “The methylation of catechol...”
R: We are very grateful for the reviewer’s suggestions and academic rigor. We have re-described “The synthesis of guaiacol from catechol...” into “The methylation of catechol...” in the revised manuscript (page 10).
- In the subsection 1.Materials, reactants and control molecules (guaiacol, veratrol, 1,2-diethoxybenzene) should be indicated.
R: Thanks a lot for the reviewer’s suggestions. In the subsection 3.1.Materials, guaiacol and veratrol have been indicated in the revised manuscript (page 9).

Reviewer 2 Report
The paper describes the O-metylation of catechol with DMC over a number of amorphous AlO4 catalysts with different Al/P ratio. The paper appears to be a follow-up to reference 17, which is from the laboratory. The study is systematic and the conclusion are mainly supported by the experimental evidence. However, the results should be discussed in the light of other catalysts tested in the reaction. Are these materials e.g. superior to microporous aluminophosphates? Moreover, detailed characterization of the catalysts with respect to surface area, chemical composition etc. to derive structure property relationships.
Some other points should also be addressed:
11) Figures 2 and 4: IR spectra are typically plotted from high to low wavenumber
22) Figure 4: Please add the spectrum of an unloaded AlPO catalysts. Rephrase ”Pyridine-loaded FT-IR spectra”.
33) The discussion on the intensities to the different IR bands is ambiguous and not in line with the spectra shown.
44) Figure 5: What is the reason for the observed activation for 2 of the 4 catalysts? Coke formation, adsorption of polar products?
55) Detailed data on the catalysts selectivities at the same conversion should be provided e.g. in separate table in addition with the real chemical composition and specific surface area. This would allow a valid comparison of the different catalysts.
66) What is subsummarized under other products in Figure 7?
77) Page 8: XRD diffraction patterns are shown rather than XRD spectra (in section 3.3 catalyst characterization).
Author Response
1) Figures 2 and 4: IR spectra are typically plotted from high to low wavenumber.
R: Thanks a lot for the reviewer’s suggestions. Figures 2 and 4 were plotted from high to low wavenumber in the revised manuscript (page 4 and page 6).
2) Figure 4: Please add the spectrum of an unloaded AlPO catalysts. Rephrase “Pyridine-loaded FT-IR spectra”.
R: Thanks a lot for the reviewer’s suggestions. According to the reviewer’s suggestion, the spectrum of an unloaded APO catalyst has been added in Figure 4a (page 6).
3) The discussion on the intensities to the different IR bands is ambiguous and not in line with the spectra shown.
R: We are very grateful for the reviewer’s suggestions and academic rigor. We have re-described “Lewis acid sites concentration sharply increased with….” into “The intensity of the band at 1448 cm−1 increased with…..” in the revised manuscript (page 5).
4) Figure 5: What is the reason for the observed activation for 2 of the 4 catalysts? Coke formation, adsorption of polar products?
R: Thanks a lot for the reviewer’s suggestions. Coke formation is a major cause of deactivation of catalysts in the O-methylation of catechol.
5) Detailed data on the catalysts selectivities at the same conversion should be provided e.g. in separate table in addition with the real chemical composition and specific surface area. This would allow a valid comparison of the different catalysts.
R: Thanks a lot for the reviewer’s suggestions. According to the reviewer’s suggestion, the detailed data on the catalysts selectivities at the same conversion have been added in Figure 5b and Figure 6b (page 7). In addition, the real chemical composition and specific surface area have been added in Table 1 (Table R2 below) and relevant descriptions into the revised manuscript (page 3).
Table R2. Physico-chemical characteristics of APO catalysts.
|
Sample |
Specific surface area (m2 g-1)a |
P/Al molar ratiob |
|
APO(0.6)-475 |
116 |
0.58 |
|
APO(0.7)-475 |
120 |
0.66 |
|
APO(0.8)-475 |
125 |
0.79 |
|
APO(0.9)-475 |
35 |
0.85 |
|
APO(0.7)-375 |
124 |
0.71 |
|
APO(0.7)-425 |
123 |
0.68 |
|
APO(0.7)-525 |
117 |
0.65 |
|
APO(0.7)-575 |
116 |
0.63 |
a Calculated using the Brunauer–Emmett–Teller (BET) method.
b Obtained by ICP.
6) What is subsummarized under other products in Figure 7?
R: We are very grateful for the reviewer’s suggestions and academic rigor. Other products in Figure 7 mainly included 3-methyl catechol and 4-methyl catechol.
7) Page 8: XRD diffraction patterns are shown rather than XRD spectra (in section 3.3 catalyst characterization).
R: We are very grateful for the reviewer’s suggestions and academic rigor. We have re-described “XRD spectra” into “XRD diffraction patterns” in section 3.3 catalyst characterization in the revised manuscript (page 9).

Round 2
Reviewer 2 Report
The paper has been improved according to the suggestions of both reviewers. It shall be published after some minor revisons.
1) Please rewrite: "Pyridine adsorbed FT-IR spectra" into FT-IR spectra after pyridine adsorption.
2) The conversion also reaches 94 %, but the slectivity is lower than the Mg-Al catalysts (Table 2). Thus, the following statement is not correct: "the catechol conversion increased and reached
a maximum of 94%, which was higher than the previously reported catalysts (Table 2). The superior catalytic activity of the APO(0.7)-475 could be explained by the presence...
How do the authors explain the lower selectvity (63 vs. 79 %!!!). Please revise the text accordingly
3) How is the performance compared to microporous aluminophosphates?
Author Response
1) Please rewrite: "Pyridine adsorbed FT-IR spectra" into FT-IR spectra after pyridine adsorption.
R: Thanks a lot for your suggestions. We have corrected “Pyridine adsorbed FT-IR spectra” into “FT-IR spectra after pyridine adsorption” in the Figure 4 in the revised manuscript (page 6).
2) The conversion also reaches 94 %, but the slectivity is lower than the Mg-Al catalysts (Table 2). Thus, the following statement is not correct: "the catechol conversion increased and reached a maximum of 94%, which was higher than the previously reported catalysts (Table 2). The superior catalytic activity of the APO(0.7)-475 could be explained by the presence...How do the authors explain the lower selectvity (63 vs. 79 %!!!). Please revise the text accordingly.
R: Thanks a lot for your suggestions. We have deleted “which was higher than the previously reported catalysts (Table 2)” and added “The APO(0.7)-475 exhibited excellent catalytic activity compared to the previously reported catalysts (Table 2)” in the revised manuscript (page 6).
3) How is the performance compared to microporous aluminophosphates?
R: Thanks a lot for your suggestions. The microporous aluminophosphates had been tested in the preliminary experiment. And it was found that the APO(0.7)-475 catalysts showed the highest catalytic performance.
